# Necrosis Links Neurodegeneration and Neuroinflammation in Neurodegenerative Disease

**DOI:** 10.3390/ijms25073636

**Published:** 2024-03-24

**Authors:** Hidenori Homma, Hikari Tanaka, Kyota Fujita, Hitoshi Okazawa

**Affiliations:** Department of Neuropathology, Medical Research Institute, Tokyo Medical and Dental University, 1-5-45, Yushima, Bunkyo-ku, Tokyo 113-8510, Japan

**Keywords:** necrosis, TRIAD, necroptosis, pyroptosis, ferroptosis, paraptosis, apoptosis, Alzheimer’s disease, Parkinson’s disease, Huntington’s disease, FTLD, ALS, HMGB1, YAP

## Abstract

The mechanisms of neuronal cell death in neurodegenerative disease remain incompletely understood, although recent studies have made significant advances. Apoptosis was previously considered to be the only mechanism of neuronal cell death in neurodegenerative diseases. However, recent findings have challenged this dogma, identifying new subtypes of necrotic neuronal cell death. The present review provides an updated summary of necrosis subtypes and discusses their potential roles in neurodegenerative cell death. Among numerous necrosis subtypes, including necroptosis, paraptosis, ferroptosis, and pyroptosis, transcriptional repression-induced atypical cell death (TRIAD) has been identified as a potential mechanism of neuronal cell death. TRIAD is induced by functional deficiency of TEAD-YAP and self-amplifies via the release of HMGB1. TRIAD is a feasible potential mechanism of neuronal cell death in Alzheimer’s disease and other neurodegenerative diseases. In addition to induction of cell death, HMGB1 released during TRIAD activates brain inflammatory responses, which is a potential link between neurodegeneration and neuroinflammation.

## 1. Introduction

In neurodegenerative diseases, neuronal cell death predominantly occurs in specific nervous systems and tracts, for example, the pyramidal system, extrapyramidal system, autonomic nervous system, and cerebellar system. Neuronal cell death occurs at specific stages of disease and, depending on the disease type, can occur following chronic or acute functional disturbances of neurons and glia in specific brain regions.

The most distinct form of selective neurodegeneration occurs in amyotrophic lateral sclerosis (ALS), which affects the pyramidal tract. The pyramidal tract is comprised of the upper and lower motor neurons, and neuronal cell death is highly selective, characterized by rapid neuronal death of motor neurons in the spinal anterior horn. Alzheimer’s disease (AD) dominantly affects the default mode network, including the dorsal medial prefrontal cortex, posterior cingulate cortex, precuneus and angular gyrus, which continuously function during the resting state of the brain [1]. Contrastingly, frontotemporal lobar dementia (FTLD) affects the salience network, comprised of the anterior insula and dorsal cingulate cortex, which responds to sensory stimuli [2]. In these chronic diseases, neuronal cell death is gradual. Further, the progression patterns of AD and FTLD are unique [3].

Despite over 30 years of intensive research, the molecular mechanisms of system-selective neurodegeneration are incompletely understood [4]. The hypotheses proposed thus far could be largely classified into (1) specific types of neurons expressing disease-causative genes, (2) specific types of neurons lacking protective mechanisms against disease toxicity, and (3) propagation of specific disease proteins in specific nervous systems. Each category of hypotheses is strongly supported by experimental evidence, although controversial findings have also been reported. No single class of hypotheses sufficiently explains all types of system-selective neurodegeneration. Therefore, system-selective neurodegeneration is likely disease-specific and modulated by combinations of multiple factors that affect specific cell types or systems.

Another issue is the purity of the pathology of human neurodegenerative diseases. Recent studies of human disease pathology have identified that some pathological features occur in multiple neurodegenerative diseases. For example, TDP43 and/or α-Synuclein are aberrantly expressed in 10–40% of postmortem AD brains, which is a well-known pathological feature of FTLD and Parkinson’s disease (PD). Such mixed pathology of multiple neurodegenerative diseases in a single patient’s brain makes the issues of system-selective neurodegeneration and the underlying neuronal cell death more complex.

In addition to the complexities of neurodegenerative disease pathology, the nature of neuronal cell death in neurodegenerative diseases remains obscure. Nearly 30 years ago, apoptosis was identified as a mechanism of cell death in neurodegenerative diseases, including AD [5]. However, the potential role of apoptotic cell death was derived from studies of primary culture neurons treated with amyloid beta (Aβ), in which DNA cleavage, a feature of apoptotic cell death, was detected by the terminal deoxynucleotidyl transferase dUTP nick end labeling (TUNEL) assay. The activation of caspases that mediate apoptosis signaling was also identified as supportive evidence for the role of apoptotic cell death in some contexts. The activation of caspase 3 to cleaves overexpressed presenilins in culture cells [6], which seemed consistent with the apoptosis hypothesis. Further studies demonstrated that other caspases are activated in AD-like conditions, supporting the apoptosis hypothesis [7,8,9,10,11]. However, the apoptosis hypothesis was disputed by other investigators, in part because the TUNEL assay produces positive signals in necrotic cells [12,13] and because apoptotic inhibitors have been unsuccessful in numerous clinical trials for human neurodegenerative diseases over the last 30 years [14]. Moreover, various types of necrosis have been identified recently [15] that have made the definition of apoptosis obscure by revealing the activation of some caspases during different forms of necrosis [14]. For example, caspase 1 in pyropotosis and caspase 9 in paraptosis [14].

Collectively, these findings suggest that necrosis, rather than apoptosis, could be the operative form of cell death in human neurodegenerative diseases. The present review discusses the nature of neuronal cell death in neurodegeneration, especially with regard to the relationship between neuronal cell death and neuroinflammation.

## 2. Variety of Necrosis Subtypes: Morphology, Biochemistry, and Signals

Robust experimental evidence now supports the role of necrosis in neuronal cell death, but the operative subtypes of necrosis remain incompletely understood. Over the past 15 years, more than ten subtypes of necrosis have been reported [16], underscoring the importance of identifying the operative subtypes of neuronal necrosis in neurodegenerative disease. Necroptosis, paraptosis, TRIAD, pyroptosis and ferroptosis have been identified as potential subtypes of neuronal necrosis and will be discussed in the chronological order of their discoveries. We emphasize the regulatory mechanisms and vacuolated cell organelles in each form of necrosis to discuss their feasibility as neuronal cell death models for neurodegenerative diseases.

### 2.1. Necroptosis

Under apoptosis-deficient conditions, necrotic cell death is induced in response to classical apoptotic stimuli such as FasL and TNFα [17,18]. Because the mechanism of cell death was changed from apoptosis to necrosis by specific apoptosis inhibitory factors, this form of cell death was termed necroptosis, as it was considered to be apoptosis-like programmed necrosis [19], which was regulated by RIP kinase [20]. The cell signaling pathways of necroptosis were further investigated [21], identifying upstream and downstream signaling molecules of RIP kinase. The first study of necroptosis in 1998 did not characterize the detailed characteristics of this process [17]. A later seminal study of necroptosis revealed a lack of chromatin condensation and apoptotic bodies, instead identifying dilatation of mitochondria and other organelles, although the organelles that dilated to or changed to cytoplasmic vacuoles were not defined [18]. Subsequent papers described the biochemical activation of autophagy in this context, identifying the presence of autophagosomes [19], although these findings did not suggest that the morphological criteria of necroptosis were homologous to autophagic cell death [22,23].

### 2.2. Paraptosis

Nonapoptotic programmed cell death induced by forced expression of the insulin-like growth factor 1 receptor intracellular domain (IGF1R-IC) was characterized by upregulation of transcriptional expression of the *caspase-9 zymogen/precursor* gene [24]. In this paper, the Bredesen group revealed that cytoplasmic vacuolation, which appeared to be derived from the endoplasmic reticulum but was not completely defined by morphological molecular markers, occurs in paraptosis [24]. In addition, mitochondrial dilatation occurs as a late event in paraptosis [24]. The same group identified that paraptosis is mediated by the MAP kinases MEK2 and JNK and that AIP-1/Alix, which interacts with the cell death-related calcium-binding protein ALG2, inhibits paraptosis, presumably by suppressing IGF1-R phosphorylation [25].

Further complicating the morphological definition of paraptosis, some reports of anti-cancer treatment claimed the existence of parapoptotic cancer cell death with morphological description but they did not completely exclude the possibility of apoptosis [26,27,28]. Further studies reporting grossly defined cancer cell death characterized by dilatation of the ER [29,30,31,32] suggested that these forms of cell death were also paraptosis, further complicating the definition of this process. These discrepancies should be addressed by further studies providing more strict definitions of necrotic cell death characterized by ER dilatation.

### 2.3. Pyroptosis

Pyroptosis was originally described as a form of apoptosis induced in macrophages by *Salmonella* invasin SipB [33,34]. In this study, the authors identified co-localization of SipB and caspase-1 in subcellular compartments and revealed direct binding between SipB and caspase-1 [34]. The interaction prompted a further study demonstrating that caspase-1 is essential for this form of cell death [34]. Morphological analyses were preliminary, and cell death was only characterized by propidium iodide (PI) [34]. Contrastingly, another group examined *Salmonella*-induced macrophage cell death, positing that this process was more consistent with necrosis than apoptosis [35]. The lack of nuclear chromatin condensation and caspase-3 activation in *Salmonella*-induced macrophage cell death were considered to be the supporting evidence for necrosis [35]. Consequently, this group proposed to term *Salmonella*-induced cell death as “pyroptosis [36]”. However, the detailed morphological features of pyroptosis, such as the presence and/or origins of cytoplasmic vacuoles, were not defined in these original studies.

### 2.4. Ferroptosis

The concept of ferroptosis originated from chemical screening to identify anti-cancer candidate drugs that were effective in Ras-mutated cancer cells, identifying erastin as a candidate compound [37,38]. Erastin-induced cell death was not characterized by nuclear fragmentation or caspase-3 activation, distinguishing this form of cell death from apoptosis [37]. Electron microscopy analysis revealed mitochondrial changes that the authors described as “shrunken mitochondria”, while cytoplasmic vacuoles were not described [37].

The Stockwell group further investigated the signaling pathways of erastin-induced cell death, suggesting that the RAS-RAF-MEK pathway activates oxidative cell death [39]. Because reactive oxygen species (ROS) promoted, while iron chelators inhibited, erastin-induced cell death, this group termed elastin-induced cell death ferroptosis [40]. Six genes were identified to be necessary for erastin-induced cell death by shRNA screening, including genes encoding ribosomal protein L8 (*RPL8*), iron response element binding protein 2 (*IREB2*), ATP synthase F0 complex subunit C3 (*ATP5G3*), citrate synthase (*CS*), tetratricopeptide repeat domain 35 (*TTC35*), and acyl-CoA synthetase family member 2 (*ACSF2*). One of the regulatory genes (*IREB2*) was consistent with the proposed iron-dependent cell death pathway, although the detailed molecular pathway remains incompletely understood [40].

### 2.5. TRIAD

TRIAD was reported as a form of neuronal cell death induced by the RNA polymerase II-specific inhibitor alpha-amanitin [41]. Our group conducted this study because transcriptional disturbance has been implicated in polyglutamine diseases, a class of neurodegenerative diseases that includes Huntington’s disease, Kennedy’s disease, spinocerebellar ataxias type 1, 2, 3, 6, 7, and 17, and dentatorubropallidoluysian atrophy. We sought to determine the effects of general transcriptional repression in neurons.

We detected an atypical form of cell death that could be classified as type 3 necrosis with cytoplasmic vacuoles, terming this form of cell death TRIAD [41]. The absence of mitochondrial cytochrome C release and genomic DNA fragmentation distinguished TRIAD from apoptosis biochemically. The presence of ER enlargement, which was confirmed by the localization of the ECFP-KDEL fusion protein, and the absence of autophagosomes, which was confirmed by the EGFP-LC3 fusion protein, distinguished TRIAD from autophagic cell death [41]. Chromatin condensation was not present, and DNA fragmentation was detected in only trace amounts [41].

The TRIAD signaling pathway has also been investigated in different model systems. The first study of TRIAD in primary mouse neurons utilized gene expression profiling to identify that YAP, the final effector molecule of the Hippo signaling pathway, potentially regulates TRIAD [41]. Drosophila genetic screening using knockdown models of 93 cell death-related genes revealed that the cell cycle regulator Plk1 and hnRNPA2B1 and hnRNPAB, components of the heterogeneous nuclear RNA complex, regulate pre-mRNA splicing, transport, and metabolism in this context, and contribute to TRIAD upstream of the Hippo pathway [42]. TRIAD is present in Huntington’s disease (HD) patients and mouse models and contributes to disease pathology in this context [43]. Biochemical analyses further confirmed the critical roles of YAP and its upstream kinase Plk1 in TRIAD and HD pathology [44].

## 3. Discrepancies and Similarities of Necrosis Subtypes

Each necrosis subtype has a specific history of discovery, defined biochemical markers, morphological features, and exclusion criteria that define the subtype (Table 1). However, similarities (100% identity in a parameter), homologies (incomplete identity in a parameter), and discrepancies between necrosis subtypes remain incompletely understood. Further compounding these discrepancies, when new studies define a role for specific necrosis subtypes in additional physiological or pathological phenomena, studies by non-original research groups sometimes have not checked the above-mentioned definitions of each necrosis subtype. The vagueness and incompleteness in defining the necrosis subtype may lead to confusion of their concepts in research fields and would disturb their clinical applications.

Further studies are essential to resolve current discrepancies in defining features of specific necrosis subtypes. We thus discuss the similarities and differences between necrosis subtypes based on findings in the literature. TRIAD, pyroptosis, paraptosis, and ferroptosis have multiple similarities and homologies (Table 1). However, unlike the other subtypes in question, RIP1/2 kinase activation does not occur in TRIAD [44] and so can be used as an exclusion factor for defining this necrosis subtype. On the other hand, exclusion criteria in each subtype to deny the other subtypes have not yet been identified for paraptosis, pyroptosis, or ferroptosis. The necrosis subtypes paraptosis, pyroptosis, and ferroptosis could therefore remain under the umbrella of necroptosis.

### 3.1. Paraptosis vs. TRIAD

Paraptosis and TRIAD can be distinguished by the effect of cycloheximide on cell death and caspase-9 activation. Like alpha-amanitin, cycloheximide induces TRIAD [41] but suppresses paraptosis [24]. However, the definition of paraptosis has been challenged by several studies. For example, glioblastoma cell death induced by NIM811, a small molecule cyclophilin-binding inhibitor, exhibited characteristics of paraptosis, but paradoxically, cycloheximide augmented cell death in this context [45]. In this study, NIM811-induced glioblastoma cell death was accompanied by dramatic ER dilation detected by ECFP-KDEL fusion protein [45], so the operative necrosis subtype is potentially TRIAD. In a comprehensive screen of necrosis subtypes in mouse models of AD and frontotemporal lobar degeneration (FTLD), TRIAD was identified as the operative necrosis subtype, and caspase-9 was not activated, further supporting this conclusion [46,47]. However, the MAPK/ERK or JNK/SAPK pathways are activated in both TRIAD and paraptosis [48,49], and morphological features such as pronounced ER dilation are highly homologous [24,41].

### 3.2. Ferroptosis vs. TRIAD

Ferroptosis signaling is dependent on the RAS-RAF-MEK pathway [39], as is TRIAD signaling [46]. Previous studies have identified iron accumulation in the aging brain [50] and in neurodegenerative diseases such as AD and PD [51], suggesting a potential role for ferroptosis in neurodegenerative diseases such as neurodegeneration with brain iron accumulation (NBIA) [52]. However, direct genetic or biochemical evidence of linkage to iron metabolism is proven in a small part of most studies like Friedreich’s ataxia and Hallervorden-Spatz syndrome, which is now known as pantothenate kinase-associated neurodegeneration (PKAN) [53,54,55]. Further, most studies implicating a role for ferroptosis in neuronal cell death did not fully assess the defining characteristics of ferroptosis in the brains of patients and mouse models of neurodegenerative diseases. Meanwhile, TRIAD necrosis has been detected in AD, PD, HD, ALS, and FTLD [43,44,46,47,56,57], while simultaneous analysis of multiple necrosis subtypes in AD and FTLD mouse models did not identify ferroptosis in cortical neurons [47,48].

### 3.3. Hypothetical Relationships between Necrosis Subtypes and Apoptosis

Based on findings in the literature, we postulate potential relationships or lack of correlation between necrosis subtypes (Figure 1a). Furthermore, we identified publications investigating each necrosis subtype, and using terms contained in the publications, we performed AI-based network analysis to calculate the distance between necrosis subtypes (Figure 1b). Though a potential caveat of this approach is the risk of obscuring the definitions of necrosis subtypes due to ambiguities in the literature and the failure of some studies to evaluate standardized criteria for necrosis subtype, the deduced relationships between necrosis subtypes were generally consistent with our human brain-based mapping (Figure 1a).

To aid in comprehending the signaling pathways in apoptosis and necrosis subtypes for general readers, we show an integrative figure summarizing the cellular pathways mentioned above (Figure 2).

## 4. Comparison of Necrosis Subtypes in Neurodegenerative Diseases

To our knowledge, no existing studies have simultaneously evaluated all forms of necrosis in neurodegenerative disease. Therefore, we performed immunostaining of multiple necrosis subtype markers in a simultaneous set of experiments (Table 2, Figure 3) to comprehensively evaluate necroptosis subtypes in mouse models of AD [48] and FTLD [47].

In this series of experiments, we used cerebral cortex tissue after ischemic injury via bilateral carotid artery stenosis as a positive control for multiple necrosis subtypes. Interestingly, immunohistochemical analyses of ischemic brains revealed that all necrosis subtypes were accompanied by MARCKS phosphorylation at Ser46 (pSer46-MARCKS). Furthermore, our previous studies revealed the link of high mobility group protein B1 (HMGB1) to pSer46-MARCKS [46,58]. HMGB1 is a protein abundant in the nucleus that is highly conserved beyond species and regulates DNA architecture [59,60]. When cells are damaged, HMGB1 is shifted to the cytoplasm and then released from cells to extracellular space [61], especially under necrosis [62]. The extracellular HMGB1 induces the activation of macrophages and microglia [61]. Also in neurons, cell signaling triggered by HMGB1 binding to TLR4 includes ERK activation leading to pSer46-MARCKS, a protein that functions in synaptic spine membrane structure [46,58] and PKC activation, resulting in the inactivation of Ku70, a protein that functions in DNA repair and maintenance [48]. The former pathway leads to neurite degeneration [46,58], and the latter pathway results in DNA damage accumulation and necrotic cell death [48].

Given that pSer46-MARCKS is a common feature of multiple necrosis subtypes [47,48] (Table 3), HMGB1-mediated neuronal cell death transmission may contribute to ischemic brain injury [46,58].

Ischemic cerebral cortex tissue was used as a positive control for markers of pyroptosis, paraptosis, and necroptosis [48]. These markers were not detectable in the AD model mice cerebral cortex tissues [48]. Only TRIAD markers (nuclear YAP disappearance and pSer46-MARCKS) were present [48]. However, all subtypes were positive for pSer46-MARCKS [48]. 5xFAD mice: B6SJL-Tg (APPSwFlLon, PSEN1*M146L*L286V)6799Vas/Mmjax, APP-KI mice: App^NL-G-F/NL-G-F^, B6 mice: C57BL/6.

## 5. Molecules Linking Diseases to Neuronal Necrosis Subtypes

In most necrosis subtypes, there are significant gaps in knowledge between the originally identified concepts and the actual use of the concept in the identification of neuronal cell death in diseases. For example, *Salmonella* invasin SipB induces pyroptosis in macrophages [33,34]. Ferroptosis is a potential therapeutic target for anti-cancer interventions, and the anti-cancer candidate drug Erastin induces cell death in Ras-mutated cancer cells [37]. Ferroptosis is mediated in part by reactive oxygen species (ROS), and iron chelation inhibits Erastin-induced cell death, which is the critical criterion for ferroptosis [40]. However, ROS as a general signal mediator also promotes cell death via ferroptosis-independent cell biological and pathological processes, and ROS-induced cell death is not sufficiently specific to ferroptosis to identify the specific operative necrosis subtypes.

It is crucial to establish direct links from disease proteins or RNA to the molecular mechanism defining specific necrosis subtypes and determine the operative subtype in neuronal cell death (not in other cell types of the brain) occurring in neurodegenerative diseases. Although studies identifying criteria for necrosis subtypes in disease states are informative, functional studies must be conducted to verify that specific necrosis subtypes are operative. For example, NLRP3-mediated inflammasome complex activation is implicated in immune cell pyroptosis. In this process, damage-associated molecular patterns (DAMPs) and/or pathogen-associated molecular patterns (PAMPs), including HMGB1 and disease proteins such as Aβ, bind to Toll-like receptor 4 (TLR4). These interactions induce phosphorylation and nuclear translocation of NF-kB, increase NLRP3 transcription, and activate the inflammasome in innate immune cells such as macrophages and microglia [63,64,65]. The active NLRP3 inflammasome complex activates caspase-1 and Gasdermin D, which induce immune cell pyroptosis of microglia [63,64,65]. In addition, concomitant ROS increases could induce ferroptosis [40].

However, because the mechanisms of inflammasome-induced pyroptosis have only been defined in immune cells, the mechanism by which inflammasome activation induces pyroptosis in neurons is currently unclear [63,64,65]. Although recent studies have investigated the role of the inflammasome in neurons [66,67,68] and provided cursory evidence for a link between inflammasome-induced pyroptosis and neuronal cell death, the operative role of Gasdermin D cleavage in neurons has not been confirmed by functional studies, leaving the neuronal pyroptosis undefined.

In TRIAD, intracellular Aβ and other disease proteins, such as huntingtin, interact with a transcription co-factor YAP, a critical molecule for cell survival, and deactivate it [44,48]. The deactivation suppresses gene expression regulated by TEAD, the target transcription factor of YAP [44]. The suppression of the YAP-TEAD axis has been implicated in cell senescence and ER dysfunction [63,64,65,69]. These comparisons of the potential molecular linkage of YAP-TEAD axis to neuronal cell death also suggest that TRIAD, rather than other necrosis subtypes, could be operative, especially in neurons of Alzheimer’s disease patients. Interestingly, HMGB1 induces TRIAD in surrounding neurons via Ku70 phosphorylation and subsequent dysfunction of the DNA damage repair system downstream of TLR4 signaling [48]. It is known that ROS induces ferroptosis [40] and that ROS levels increase after induction of TLR signaling [70], while the question of whether ferroptosis can also be induced in neurons via TLR4 has not been examined so far.

We show a summary of the knowledge on the molecular link of necrosis subtype to diseases (Table 4).

## 6. TRIAD, PANTHOS, and PAAS Could Be Multiple Sides of the Same Necrosis

Following the initial discovery of alpha-amanitin-induced TRIAD necrosis in cultured primary neurons influenced by the presence of different YAP isoforms [41], a similar phenotype mediated by YAP inactivation has been detected in patients with Huntington’s disease [43] and in mouse models of the disease [44]. YAP inactivation has been found to occur through interaction with the abnormal huntingtin protein (Htt) and is associated with disease pathology [44]. Interestingly, also in AD, TRIAD necrosis was detected in neurons where accumulated intracellular Aβ interacts and deactivates YAP at the preclinical stage of Alzheimer’s disease prior to the development of extracellular Aβ plaques [46]. Our recent study, in fact, demonstrated that AAV-based gene therapy rescues intracellular Aβ-induced suppression of nuclear YAP and prevents subsequent initiation of TRIAD signaling. Further, interaction with intracellular Aβ [46] or treatment with an anti-HMGB1 antibody, which inhibits paracrine HMGB1-mediated induction of TRIAD in proximal neurons, significantly inhibited disease progression in AD mouse models [46,47,56].

Interestingly, 2 years after this study, two independent groups identified homologous and potentially similar pathologies to TRIAD in the PANTHOS and PAAS neurodegenerative processes (https://www.alzforum.org/news/research-news/dystrophic-neurites-dampen-long-range-neuronal-signaling#comment-47951 (accessed on 21 March 2024)) (Figure 4). The Grutzendler group investigated the mechanism of cognitive decline in AD and revealed swelling of axons (axonal spheroids) around Aβ plaques, which they named plaque-associated axonal spheroids (PAAS) and was related to the synaptic dysfunction [71]. Regarding the mechanism, they demonstrated that LAMP1-positive vesicles formed by the autophagy-lysosomal degradation pathway accumulated and caused axonal spheroid formation. The Grutzendler study also posited that a risk factor, gene product PLD3, which is located in the ER, endosomes, and lysosomes, is essential for the accumulation of LAMP1-positive vesicles [71]. The Grutzendler study suggested that these structures are likely derived from degenerative neurites originating from other neurons, such as those of the contralateral hemisphere. It is of note that they showed PAASs around very small plaques of a single cell size, suggesting PAAS is made around a single cell necrosis due to intracellular Aβ accumulation at the early stage of plaque formation. Their finding is also important because axonal spheroids form in multiple neurodegenerative diseases in addition to Alzheimer’s disease.

A study from the Nixon group suggested that the cytoplasmic regions filled with autophagosome-lysosomes protruded from a dying neuron with intracellular Aβ accumulation, a process termed poisonous anthos (=flower) (PANTHOS) [72]. The Nixon group was originally interested in autophagy function in AD and monitored changes of a neuron-specific transgenic mRFP-eGFP-LC3 probe that could monitor pH and distinguish autophagosomes and autolysosomes [72]. Furthermore, they used cathepsin D together with an mRFP-eGFP-LC3 probe further to investigate the acidification of autolysosomes [72]. Their results indicated that acidification deficiency of autolysosomes occurred in neurons before the formation of extracellular Aβ plaques and that such neurons showed plasma membrane blebbing fulfilled with autophagic vacuoles (AV) [72]. Intriguingly, the AV-filled structures are morphologically similar to those reported by the Grutzendler group. Moreover, in our report, prior to these two groups that identified the occurrence of TRIAD at the preclinical stage of AD pathology, we detected degenerative neurites with robust autophagosome accumulation [46]. Together, these findings suggest that TRIAD, PANTHOS, and PAAS necrosis are likely to be the same necrosis subtype defined by different but overlapping criteria (Figure 4). Further, the “trinity” pathology is consistent with the classic pathology of Aβ plaques described in Greenfield’s Neuropathology Textbook [73].

## 7. TRIAD and Other Necrosis Subtypes in the Pathology of Parkinson’s Disease

Similarly to the case of AD, DNA fragmentation observed in a TUNEL assay has been a focus of discussion in PD, and there are some proofs in human pathology that apoptosis does not exist in neurons of PD/DLB [74]. After 20 years of such discussion, recent investigations now suggest the involvement of necrosis subtypes in PD/DLB.

TRIAD is implicated in the pathology of Parkinson’s disease (PD) and related dementia with Lewy bodies (DLB) [56]. Our group performed a comprehensive phosphoproteome analysis of DLB and AD postmortem brains by mass spectrometry and found that pSer46-MARCKS was commonly increased in some brain regions [56]. Immunohistochemistry and western blots of human DLB brain samples supported the elevation of pSer46-MARCKS in neurons and neurites, suggesting that TRIAD occurs in alpha-synuclein-associated neurodegenerative diseases [56]. Our group further investigated the initial timing of pSer46-MARCKS emergence in the brain of a PD/DLB mouse model (normal human α-Syn-BAC-Tg mice [75]) and found that the signal of pSer46-MARCKS in neurons and neurites were increased at 1 month of age before the appearance of pSer129-α-Synuclein-reactive inclusions in neurons at 24 months of age in multiple brain areas, including the olfactory bulb [56], where the α-Synuclein aggregates occur at the earliest timing [76]. These notions suggested that TRIAD could be a domain of pre-aggregation pathology at an ultra-early stage not only in AD but also in PD/DLB [56].

Necroptosis is also implicated in PD/DLB by the results that nectrostatin, an inhibitor of necroptosis, partially (nearly 30%) suppressed the MPTP-induced model of PD [77], while the other study suggested the toxicity of necrostatin on cell death in primary fibroblasts from PD patients with and without the G2019S *leucine-rich repeat kinase 2* (*LRRK2*) mutation and in rotenone-treated cells (SH-SY5Y and fibroblasts) [78].

Ferroptosis is the third candidate of necrosis subtype for PD/DLB [79]. Iron accumulation in affected neurons of PD patients suggests ferroptosis may occur in PD [80], and in vitro cell lines treated with MPP+ were sensitive to a ferroptosis inhibitor [81,82]. Intriguingly, the ferroptosis-like necrosis in the PD model was atypical and triggered by the activation of the PKC-ERK-MEK signaling pathway [81], similar to TRIAD [46,48]. This finding might be important for defining the necrosis subtype in PD/DLB, considering the biochemical and morphological homologies between TRIAD and ferroptosis (Figure 1). Though the distance between TRIAD and ferroptosis is long in GO analysis (Figure 1b), the signaling pathway identity might indicate that different groups observed the same necrosis subtype in PD/FTLD pathologies.

## 8. Protein Aggregation and Necrosis Subtypes

Protein aggregation has been a central dogma in various neurodegenerative diseases, whereas recent data from multiple groups suggested the existence of pre-aggregation pathology [83,84,85,86]. For instance, in addition to the recently reported neuronal changes before the appearance of extracellular Aβ plaques [46,71,72], protein–protein interactome analyses revealed pathological roles of pre-aggregated soluble disease proteins interacting with physiological proteins [87,88,89,90]. Hence, cell death could occur from the early stage of pathology from the aspect of disease protein aggregation. Here, we intended to position the timing of necrosis subtypes at the disease protein aggregation stages on the basis of molecular signaling that could possibly be induced by each aggregation state of disease proteins (Figure 5).

Soluble but not insoluble Htt protein can interact with Ku70 to impair the DNA damage repair function in non-homologous end joining repairing DNA double-strand breaks in neurons [90], and the consequently accumulated DNA damage in HD leads to TRIAD [43,44], suggesting that TRIAD occurs in the pre-aggregation stage. The idea of the timing of TRIAD is further supported by the chronological observation of two AD mouse models and by comparative observation of human MCI and AD patients [46]. In the cases of four FTLD mouse models, TRIAD was observed during the developmental stage when disease protein aggregation was not observed in neurons [47]. Moreover, HMGB1-mediated reproduction of TRIAD continues to the late stage of AD pathology after the emergence of extracellular Aβ plaques [48].

In the case of ferroptosis, iron accumulation in neurons is the trigger of cell death. MRIs of normal-aged people indicated that iron accumulation might occur before the emergence of protein aggregation or some other neurodegenerative changes [50]. However, the cause–result relationship of protein aggregation and iron accumulation remains obscure despite an enormous amount of publications [52]. Though a gene related to iron metabolism called *hemostatic iron regulator* (*HFE*), whose mutations cause hemochromatosis, has been implicated as a risk factor for AD, the results published so far are controversial [91,92]. There is no report of chronological analysis of ferroptosis in animal models of neurodegenerative diseases to the best of our knowledge. Collectively, the timing of ferroptosis, if any, is presumably at the early stage but not definite during the course of protein aggregation.

Necroptosis has been analyzed in PD and AD models. In the PD mouse model treated with the neurotoxin MPTP, an inhibitor had a therapeutic effect on the neurodegeneration of dopaminergic neurons [77]. However, the report added no information to the relationship between necroptosis and protein aggregation because the MPTP mouse model does not accompany α-synuclein protein aggregation. In human postmortem AD brains, according to the report by Caccmo and colleagues, RIPK activation was associated with intracellular tau accumulation, suggesting that necroptosis is an event at the late stage of protein aggregation [93].

The Dawson group revealed that loss of parkin activity in mouse and human dopaminergic neurons induced spontaneous neuronal NLRP3 inflammasome assembly that leads to neuronal death of dopaminergic neurons [67]. The result suggested that pyroptosis, which is characterized by inflammasome assembly [94], could occur from the early stage of protein aggregation, at least in familial PD with the *Parkin* gene mutation, while the verification of the cell death timing with regard to protein aggregation has not been performed.

Collectively, necrosis could initiate before protein aggregation or continue from early to late stages during the course of protein aggregation, yet a detailed chronological analysis has not been performed except in the case of TRIAD (Figure 5).

## 9. HMGB1 Released from Necrotic Neurons Induces Neuroinflammation

HMGB1 binds to TLRs, activates the inflammasome, and upregulates the expression of pro-IL-1beta and pro-IL-18 in brain microglia [95]. Synergistically, Aβ released from necrotic neurons activates the NLRP3 inflammasome in innate immune cells and activates caspase-1, which cleaves pro-IL-1beta and pro-IL-18 into their active forms [96]. HMGB1, Aβ and Tau are representative DAMPs (also considered as PAMPs and SASP) that are released by necrotic neurons exhibiting intracellular Aβ accumulation as in the TRIAD pathway [46,56]. Together, these findings suggest that neuronal necrosis and the release of HMGB1 and other DAMPs from necrotic neurons trigger brain inflammation (Figure 6).

Microglial activation is considered to be a key event of pathological neuroinflammation in AD and other neurodegenerative diseases [96,97]. However, activated microglia can also have beneficial effects, for example, disease-associated microglia (DAM) that could be classified as M2 microglia and phagocytose extracellular Aβ aggregates [98]. Contrastingly, neuroinflammation induced by HMGB1-activated or Tau-activated M1 microglia damages neurons via pro-inflammatory cytokines like TNFα, INFγ and IL-1β [99,100] and paradoxically increases the accumulation of disease protein aggregates in the brain [48,101]. In Tau-activated microglia, PQBP1 functions as an intracellular receptor for Tau, and the PQBP1-cGAS-STING pathway mediates cell signaling [99]. The context and subtype-dependent protective and deleterious roles of microglia in neurodegeneration have been reviewed in depth elsewhere [96,97].

**Figure 6 ijms-25-03636-f006:**
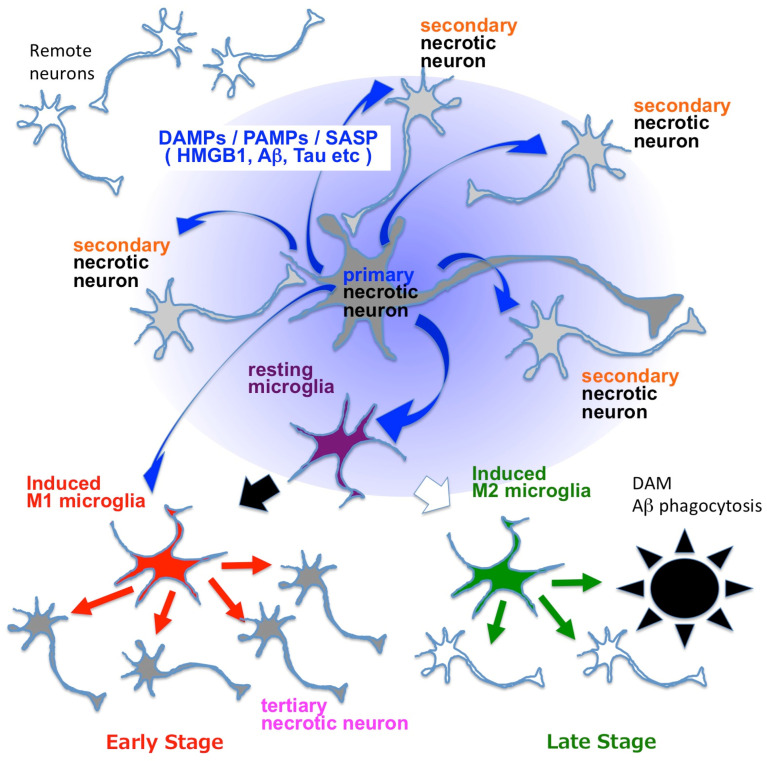
**Mechanisms linking necrosis and neuroinflammation in neurodegenerative diseases.** Blue arrows indicate DAMPs such as HMGB1, Aβ and Tau, which are also known as PAMPs or SASP. DAMPs activate resting microglia and change them to M1 or M2 microglia. M1 microglia secrete pro-inflammatory cytokines such as TNFα, INFγ, IL-6 and IL-1β (red arrows), while M2 microglia secrete anti-inflammatory cytokines like IL-10, IL-13, IL-14 and TGFβ (green arrows). Blue arrows indicate DAMPs/PAMPs/SASP, among which HMGB1 is considered as a representative molecule [46,48,58]. DAM: disease-associated microglia [98]. Primary necrosis induces secondary necrosis via extracellularly released HMGB1, as revealed in the case of TRIAD [46,48,58]. HMGB1 and other DAMPs/PAMPs/SASP molecules that are released from neurons under primary and secondary necrosis activate microglia and induce brain inflammation, which induces tertiary necrosis of neurons.

**Table 4 ijms-25-03636-t004:** **Summary of the molecular changes and suspected necrosis subtypes in diseases.** Investigated objects, morphological and biochemical phenotypes, and necrosis subtypes are summarized for each disease with references.

Disease Name	Investigated Cells or Animals	Morphological & Biochemical Phenotypes	Necrosis Subtype	Reference
Alzheimer’s disease	mouse model (5xFAD mouse, APP-KI mouse), human iPSC-derived neuron, human postmortem AD brain	ER enlargement, nuclear reduction and cytoplasmic translocation of YAP, dysfunction of TEAD-YAP transcription, MARCKS phosphorylation	TRIAD	[46,48,58]
	culture cell, organotypic slice culture	iron accumulation, RAS-RAF-MEK pathway, ROS	ferroptosis	[50,51]
Parkinson’s disease	cell culture, mouse model (a-Syn-BAC-Tg/GBA-hetero-KO mouse), human postmortem PD brain, human iPSC-derived neuron	MARCKS phosphorylation, activation of RAS-RAF-MEK pathway	TRIAD	[51,52,56]
	cell culture, organotypic slice culture, mouse model (MPTP-treated mouse) human MRI, human postmortem PD brain	iron accumulation, neuromelanin accumulation, ROS	ferroptosis	[80,81,82]
	cell culture, mouse model (MPTP-treated mouse), human iPSC neuron	RIP kinase phosphorylation/activation, MLKL phosphorylation	necroptosis	[77,78]
frontotemporal lober degeneration	mouse model (mutant PGRN-KI, mutant TDP43-KI, mutant VCP-KI, and mutant CHMP2B-KI) human iPSC-derived neuron, human postmortem FTLD brain	ER enlargement, nuclear reduction and cytoplasmic translocation of YAP, dysfunction of TEAD-YAP transcription, MARCKS phosphorylation	TRIAD	[47]
Huntington’s disease	cell culture, drosophila model, mouse model (R6/2 mouse, HdhQ111 knock-in mouse), human postmortem HD brain	ER enlargement, nuclear reduction and cytoplasmic translocation of YAP, dysfunction of TEAD-YAP transcription, MARCKS phosphorylation	TRIAD	[41,42,43,44]
amyotrophic lateral sclerosis	mouse model (G93ASOD1 transgenic mice)	YAPdeltaC decrease, p73 decrease	TRIAD	[57]
PKAN (Friedreich’s ataxia and Hallervorden-Spatz syndrome)	yeast cell, cell culture, mouse model, human postmortem brain	iron accumulation, ROS	ferroptosis	[53,54,55]
brain ischemia	mouse model of transient focal cerebral ischemia	MARCKS phosphorylation, nuclear reduction and cytoplasmic translocation of YAP	TRIAD	[46,58]
		RIP kinase phosphorylation/activation, MLKL phosphorylation, autophagosome	necroptosis	[19,46,58]
		caspase 1 activation	pyroptosis	[46,58]
		caspase 9 activation	paraptosis	[46,58]
cancer	cell culture, xenograft model, zebrafish model, 3D cultures	caspase 9 activation	paraptosis	[26,27,28,29,30,31,32]
	cell culture	activation of RAS-RAF-MEK pathway, increase of ROS	ferroptosis	[37,38,39,45]
Salmonella infection	cell culture	Binding between SipB and Caspase-1, caspase-1 activation, PARP activation	pyroptosis	[33,34,35,36]

## 10. Conclusions

In addition to TRIAD, additional studies suggest that necroptosis, pyroptosis, and paraptosis are present in the brains of neurodegenerative disease model mice. However, the operative necrosis subtypes for neuronal cell death in neurodegenerative disease remain incompletely understood. Markers of specific necrosis subtypes were simultaneously measured in the 5XFAD and APP-KI mouse models of AD using immunostaining, suggesting that TRIAD or TRIAD-related necrosis subtypes such as pyroptosis or paraptosis could be the operative necrosis subtypes in this context.

Necrotic neuronal cell death could be the basis for a new class of therapeutics for neurodegenerative disease. Although recent clinical tests of candidate drugs have recruited early-stage patients, the timing of neuronal cell death could be even earlier than previously expected. Therefore, candidate drugs that could inhibit early-stage necrotic neuronal cell death could prevent disease progression. Targeting necrotic neuronal cell death requires that the operative necrosis subtype(s) be identified so that these pathways can be targeted in new therapeutic interventions.

## Figures and Tables

**Figure 1 ijms-25-03636-f001:**
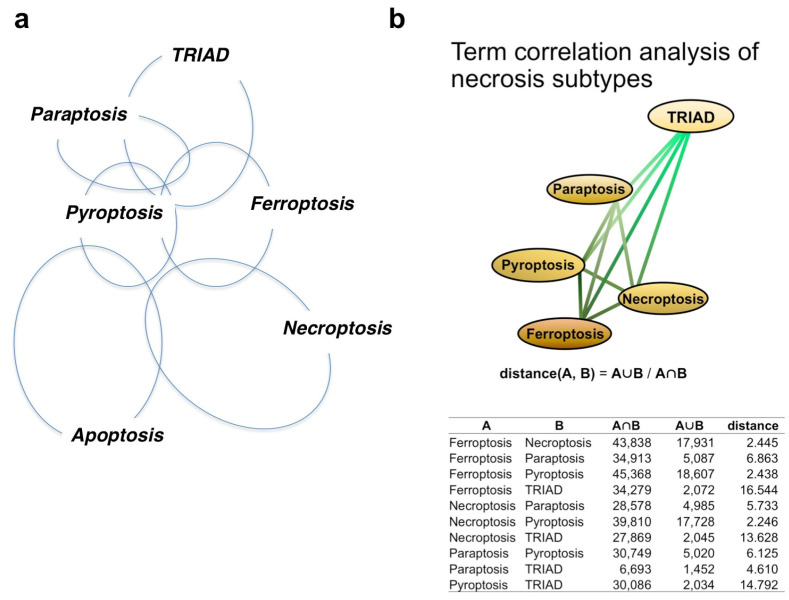
**Relationship of necrosis subtypes and apoptosis.** (**a**) Relationship of necrosis subtypes and apoptosis judged by the human brain. The pairwise similarities between necrosis subtypes were determined using information from original papers, and the relationships between five necrosis subtypes and apoptosis were reconstructed. (**b**) Relationships between subtypes, as determined by term correlation analysis. Pairwise similarities between necrosis subtypes were determined by term correlation analysis derived from published papers, and the relationships between the five assessed necrosis subtypes were reconstructed by defining the pairwise distance between subtypes as being reciprocal to the pairwise ratio intersection group and union group.

**Figure 2 ijms-25-03636-f002:**
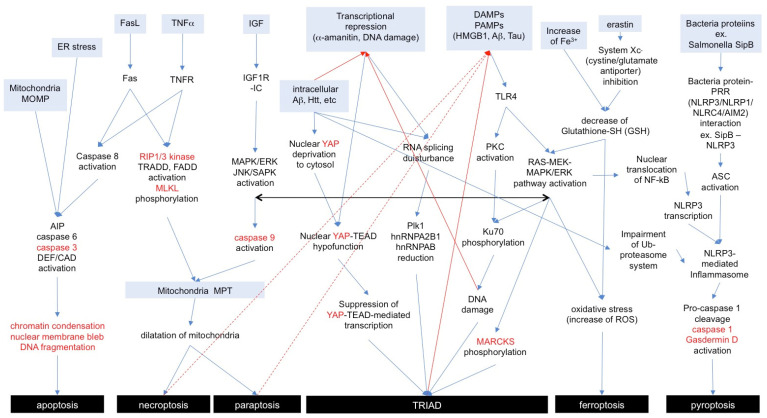
**Signaling pathways of necrosis subtypes and apoptosis.** Anterograde or retrograde orientations of the signaling pathway are indicated with blue or red arrows. Marker proteins for each necrosis subtype are shown in red letters. A similar cell signaling among subtypes is indicated with a black arrow. AIP: apoptosis-inducing factor, DEF: DNA fragmentation factor, CAD: caspase-activated DNase, MOMP: mitochondrial outer membrane permeabilization, MPT: membrane permeability transition, PRR: pattern recognition receptor, NLR: nucleotide-binding domain and leucin-rich repeat, NLRP3: NLR family PYD domain containing protein 3, NLRP1: NLR family PYD domain containing protein 1, NLRC4: NLR family CARD domain containing protein 4, AIM2: absent-in-melanoma-2, ASC: apoptosis-associated speckle-like protein containing a CARD.

**Figure 3 ijms-25-03636-f003:**
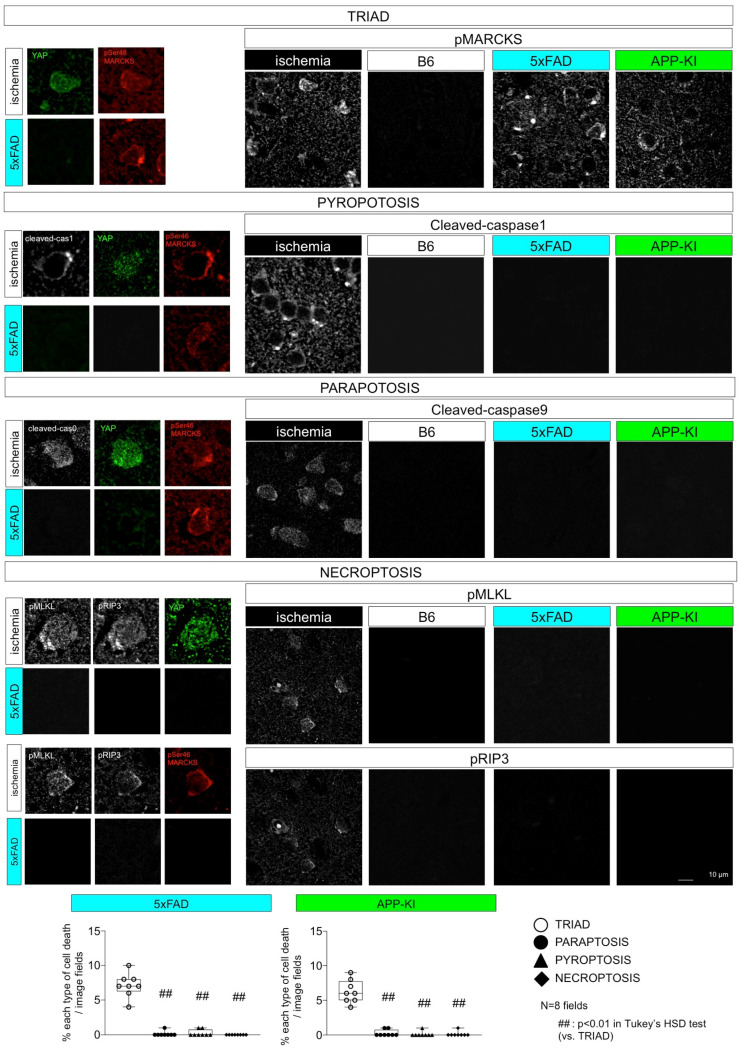
**Evaluation of various necrosis subtypes in AD pathology.** Simultaneous measurement of necrosis subtype markers in AD model mice suggests TRIAD as the operative necrosis subtype in neurodegeneration.

**Figure 4 ijms-25-03636-f004:**
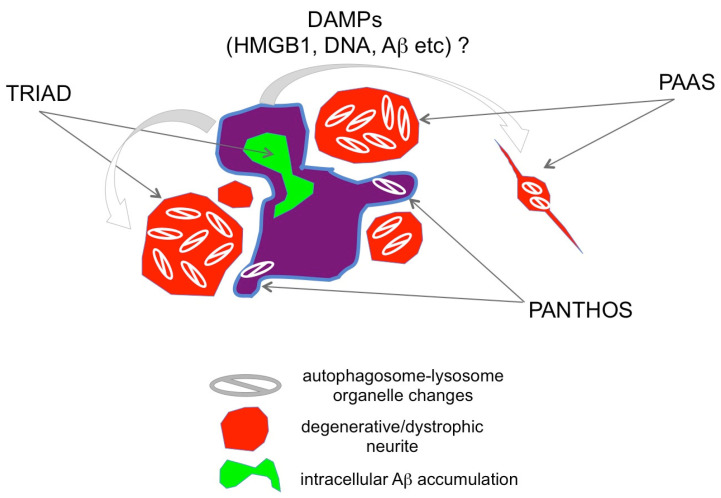
**Relationship between TRIAD, PANTHOS, and PAAS.** TRIAD, PANTHOS, and PAAS potentially reflect three aspects of a single phenomenon, characterized by degenerative/dystrophic neurites (red) surrounding and proximal to necrotic neurons (purple). Necrosis occurs due to intra-neuronal Aβ accumulation (green). Necrotic neurites contain tightly packed autophagosome-lysosomes (https://www.alzforum.org/news/research-news/dystrophic-neurites-dampen-long-range-neuronal-signaling#comment-47951 (accessed on 21 March 2024).

**Figure 5 ijms-25-03636-f005:**
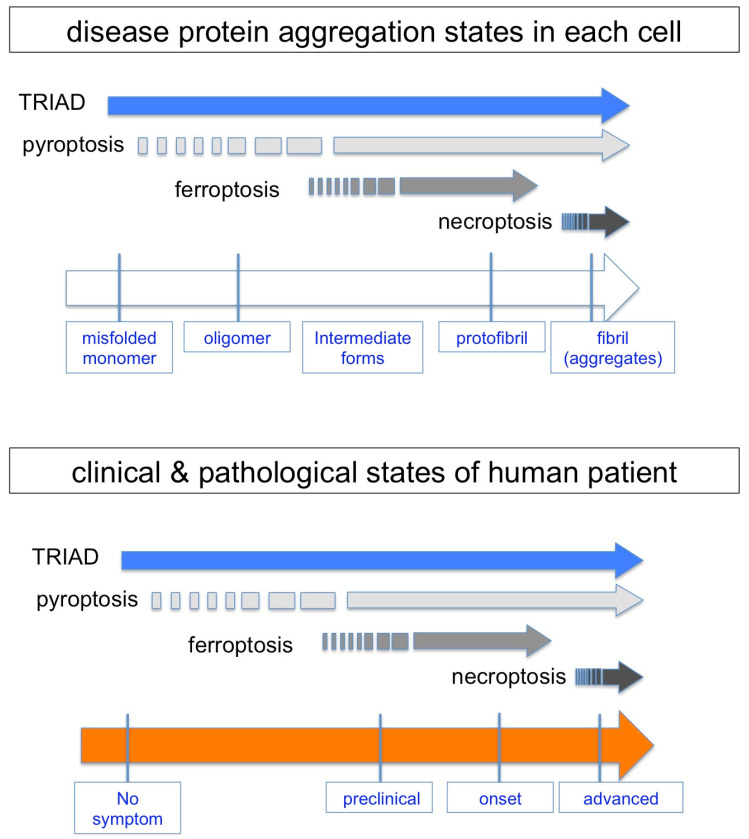
**Hypothetical emergence of necrosis subtypes during protein aggregation and clinicopathological states**. Reported results about the initial emergence and continuity of each necrosis subtype are summarized from the aspect of disease protein aggregation states in each cell. A relationship between necrosis subtypes and clinicopathological states of human patients is expected on the basis of stochastic changes of pathological neurons with different protein aggregation states.

**Table 1 ijms-25-03636-t001:** **Characteristics of necrosis subtypes.** Characteristics of necrosis subtypes are summarized with regard to original discovery, original cell type, biochemical features, morphological features, and exclusion criteria that should not be observed in the necrosis subtype. Note: KDEL, ECFP-KDEL fusion protein; MLKL, mixed lineage kinase domain-like protein.

	Origin of Discovery	Original Cell Type	Biochemical Features	Morphological Features	Exclusion Criteria
** *Necroptosis* **	FasL- and TNFa-induced cell death in apoptosis-deficient conditions	L929 cells with zVAD.fmk Jurkat-derived cell line that is deficient in caspase-8 (JB-6)	RIP kinase phosphorylation/activation MLKL phosphorylation	mitochondria dilatation cytoplasmic vacuole of unknown origin	Not apoptosis No DNA ladder No PARP cleavage
** *Paraptosis* **	Cell death induced by forced expression of intracellular domain of insulin-like growth factor 1 receptor (IGF1R-IC)	293T cells 293, MCF-7, Cos-7, and primary mouse embryonic fibroblasts	Inhibited by actinomycin D and by cycloheximide Caspase 9 activation	Cytoplasmic vacuole derived from endoplasmic reticulum Mitochondria late swelling	No autophagy activation No caspase 3 activation
** *Pyroptosis* **	Cell death induced by *Salmonella* invasin SipB in macrophages	Macrophage	Binding between SipB and Caspase-1 Caspase-1 activation PARP activation Inhibited by Glycine	Non-apoptotic in PI stain	No nuclear chromatin condensation No caspase-3 activation
** *Ferroptosis* **	Chemical screening to find anti-cancer candidate drugs that are effective on Ras-mutated cancer cells	*RAS* mutated cancer cells	Erastin-induced cell death of RAS mutated cancer cells RAS-RAF-MEK pathway reactive oxygen species (ROS) iron chelator inhibits the cell death	Shrunken mitochondria	No nuclear fragmentation No caspase-3 activation
** *TRIAD* **	Neuronal cell death induced by RNA polymerase II-specific inhibitor, alpha-amanitin	Neuron	Cytoplasmic translocation of YAP MARCKS phosphorylation Induced by actinomycin D Possibly related to BRAF-ERK-MAPK	Cytoplasmic vacuole of endoplasmic reticulum confirmed by KDEL Mitochondrial late swelling Weak DAPI stain	No chromatin condensation No DNA fragmentation No autophagy activation No caspase 3 or 7 activation

**Table 2 ijms-25-03636-t002:** **Criteria used for simultaneous analysis of necrosis subtypes in AD and FTLD mouse models.** A circle indicates the positive criteria for the necrosis subtype, and a cross indicates the negative criteria for the necrosis subtype.

	pS46-MARCKS	Nuclear YAP Reduction	pMLKL & pRIP	Cleaved Caspase 1	Cleaved Caspase 9
TRIAD	**◯**	**◯**	✖		
necroptosis			**◯**		
pyroptosis				**◯**	
paraptosis					**◯**

**Table 3 ijms-25-03636-t003:** **Results of simultaneous analyses in ischemic brain tissue.** A circle indicates a positive finding in the actual experiment for the necrosis subtype, and a cross indicates a negative finding for the necrosis subtype. A triangle indicates a positive finding in a small portion of cells under the necrosis subtype.

	pS46-MARCKS	Nuclear YAP Reduction	pMLKL & pRIP	Cleaved Caspase 1	Cleaved Caspase 9
TRIAD	◯	◯	✖	✖	✖
necroptosis	◯	✖	◯		
pyroptosis	◯	△		◯	
paraptosis	◯	✖			◯

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
