# Peer review of "Necrosis Links Neurodegeneration and Neuroinflammation in Neurodegenerative Disease"

_ijms, 2024, doi:10.3390/ijms25073636_

Round 1
Reviewer 1 Report
Comments and Suggestions for Authors
In manuscript entitled “Necrosis links neurodegeneration and neuroinflammation in neurodegenerative disease” by Hidenori Homma et al., describe different types of cell death occurring during neurodegeneration. As authors noticed knowing the pathway leading to neuronal cell death is important as it can be proposed as a therapeutic target in neurodegeneration.
I have some major comments and a list of minor ones.
I don't really understand the title of this paper. Is it about neurodegeneration as a result of the combination of necrosis and inflammation of the nervous system? I'm also having trouble understanding the assignment of a given text to a given subsection. If we look at Chapter 2 “Apoptosis and necrosis” and then Chapter 3 “Variety of necrosis:…” it seems that necrosis is not needed in Chapter 2, since the whole of Chapter 3 is devoted to it.
There is a nice review https://doi.org/10.1038/s12276-023-01078-x describing different types of programmed cell death. Here authors wanted to draw our attention to inflammation, but this part is not well presented.
There is a group of diseases called neurodegeneration with brain iron accumulation (NBIA), and this group includes more than Hallervorden-Spatz syndrome. It should also be noted that this disease is now called PKAN for ethical reasons.
A review paper is not an original paper in which results are included. If you want to present your results, you have to publish them first. I do not know why a different protein is shown for each type of necrosis; this is an experiment which, once in the wrong place, is also useless.
The conclusion chapter should summarize what has been written, include conclusions and not introduce new information, for example, about gene therapy. Further I think that the authors should work a little on the structure of the text. Make sure that the information is not repeated, the structure is well thought out and the title fits the content of the chapter.
The problems with the text are shortcuts and sometimes the jargon we use in the laboratory. The publication must be written in correct language. An example is the title of chapter three, which suggests that necrosis has morphology. The description of morphology can at most refer to cells or organelles but not to process.
I do not understand what it means that the regulatory genes (i.e. which ones) are consistent with the pathway (lines151-153)
Similarly it is with the ER morphology study described as: monitoring KDEL (in text lines 165, 209 and table 1). KDEL is the sequence of amino acid residues present at the C-terminus of ER proteins, which determines their return from the Golgi apparatus. The study uses a fusion of the green fluorescent protein GFP with the above-mentioned sequence to visualise the ER in a fluorescence microscope.
The same kind of error in a text “the absence of autophagosomes, which was confirmed by LC3.” How was this done? LC3 is a protein necessary for autophagosomes formation and present in autophagosomes membranes. So GFP-LC3 fusion can be used to observe microscopically formation of GFP-LC3 puncta. Also the lipidation of LC3 as a marker of autophagy induction can be monitored by Western blot. So, I don’t understand the real meaning of authors sentence.
Disease does not have neurons (line 309) but patients with disease.
Line 312 What question has to be examined?
Lines 314-318 There is a repetition of information from line 155. Limit it.
I don’t understand the sentence in lines 186-188. Please clarify.
In the text in several places the genes and proteins are mixed. The best example is the sentence in line 146 “genes were identified… including ribosomal protein L8…” should be including genes encoding ribosomal protein L8, iron response….etc.
I don't like the use of the term homology to refer to the process, a term used several times by the authors. I don't seem to understand it.
What are circles, triangles and crosses in Table 2 and 3. I don’t understand the tables.
The sentence in lines 258-262 is too long and difficult to understand, it should be simplified. Please also clarify the relationship between HMGB1 and cell death. The information about HMGB1 itself and its role in inflammation is missing.
There are some typos (lines 55, 84), unnecessary underlying like in lines 33-34.
Author Response
Reviewer 1
In manuscript entitled “Necrosis links neurodegeneration and neuroinflammation in neurodegenerative disease” by Hidenori Homma et al., describe different types of cell death occurring during neurodegeneration. As authors noticed knowing the pathway leading to neuronal cell death is important as it can be proposed as a therapeutic target in neurodegeneration.
>>> Thank you very much for kind evaluation of our manuscript.
I have some major comments and a list of minor ones.
I don't really understand the title of this paper. Is it about neurodegeneration as a result of the combination of necrosis and inflammation of the nervous system? I'm also having trouble understanding the assignment of a given text to a given subsection. If we look at Chapter 2 “Apoptosis and necrosis” and then Chapter 3 “Variety of necrosis:…” it seems that necrosis is not needed in Chapter 2, since the whole of Chapter 3 is devoted to it
>>> We are very sorry, but it is puzzling for us whether the yellow-marked comment is on necrosis or apoptosis. From the line of logic, we believe that the reviewer 1 is saying that descriptions about apoptosis are not needed in Chapter 2. Following this criticism of reviewer 1, we deleted Chapter 2 and integrated the content into Introduction. We believe that matching of the title of this paper with its content becomes better by this change.
There is a nice review https://doi.org/10.1038/s12276-023-01078-x describing different types of programmed cell death. Here authors wanted to draw our attention to inflammation, but this part is not well presented.
>>> We appreciate introducing a nice review paper on the history of discovery of various types of cell death forms. We referred this paper in Introduction and cast attention to the history.
There is a group of diseases called neurodegeneration with brain iron accumulation (NBIA), and this group includes more than Hallervorden-Spatz syndrome. It should also be noted that this disease is now called PKAN for ethical reasons.
>>> We appreciate the comment of reviewer 1. We improved the corresponding part in our manuscript.
A review paper is not an original paper in which results are included. If you want to present your results, you have to publish them first. I do not know why a different protein is shown for each type of necrosis; this is an experiment which, once in the wrong place, is also useless.
>>> We respect but disagree with the comment. We wrote the paper on the basis of published results. The GO-term analysis is within comparison of multiple published results from other groups not only from ours. Also GO-term analysis is no more than comparison of previous knowledge, just like manual comparison of multiple published papers in ordinary review paper.
The conclusion chapter should summarize what has been written, include conclusions and not introduce new information, for example, about gene therapy. Further I think that the authors should work a little on the structure of the text. Make sure that the information is not repeated, the structure is well thought out and the title fits the content of the chapter.
>>> We appreciate the kind advice that new information should better removed from Conclusion part. We deleted this part from Conclusion and moved it to the section “TRIAD, PANTHOS, and PAAS could be multiple sides of the same necrosis”.
The problems with the text are shortcuts and sometimes the jargon we use in the laboratory. The publication must be written in correct language. An example is the title of chapter three, which suggests that necrosis has morphology. The description of morphology can at most refer to cells or organelles but not to process.
>>> We appreciate the careful checking by the reviewer 1 that pointed out our careless use of the term “morphology”. We corrected the phrase “the detailed cell morphology characteristic of this process” to “the detailed characteristic of this process”. We think this mistake occurred during English editing by professional editors.
I do not understand what it means that the regulatory genes (i.e. which ones) are consistent with the pathway (lines151-153)
>>> We thank the comment. We corrected this part.
Similarly it is with the ER morphology study described as: monitoring KDEL (in text lines 165, 209 and table 1). KDEL is the sequence of amino acid residues present at the C-terminus of ER proteins, which determines their return from the Golgi apparatus. The study uses a fusion of the green fluorescent protein GFP with the above-mentioned sequence to visualise the ER in a fluorescence microscope.
>>> I think the comment of the reviewer 1 is the copy of description about KDEL in Wikipedia (https://en.wikipedia.org/wiki/KDEL_(amino_acid_sequence)). We imagine that the reviewer is recommending to say “KDEL-GFP fusion protein” rather than “KDEL”, so we changed these parts in text and Table 1 according to this idea. Actually, we used ECFP-KDEL fusion protein in our original paper (Hoshino et al, The Journal of Cell Biology 2006, https://rupress.org/jcb/article/172/4/589/44169/Transcriptional-repression-induces-a-slowly). Therefore, we replaced KDEL to ECFP-KDEL fusion protein in the text. We also added note to Table 1.
The same kind of error in a text “the absence of autophagosomes, which was confirmed by LC3.” How was this done? LC3 is a protein necessary for autophagosomes formation and present in autophagosomes membranes. So GFP-LC3 fusion can be used to observe microscopically formation of GFP-LC3 puncta. Also the lipidation of LC3 as a marker of autophagy induction can be monitored by Western blot. So, I don’t understand the real meaning of authors sentence.
>>> We also changed “LC3” to “EGFP-LC3 fusion protein” as we described previously (Hoshino et al. JCB 2006).
Disease does not have neurons (line 309) but patients with disease.
>>> We thank the comment on our mistake.
Line 312 What question has to be examined?
>>> We do not think that native speakers cannot understand this phrase. They (English editors) say that it means “. Increased ROS levels by TLR4 signaling could also induce ferroptosis” is the question.
Lines 314-318 There is a repetition of information from line 155. Limit it.
>>> We really think that this repeated description (lines 314-316) is necessary for readers to understand the flow to “a similar phenotype mediated by YAP deactivation…..huntingtin (Htt) protein”, and our professional English editors agree with this idea.
I don’t understand the sentence in lines 186-188. Please clarify.
>>> We again asked our English editors whether the sentence is unclear, and they said “No”and “This is clear and easy to understand”. This sentence is just mentioning about general risks by “The vagueness and incompleteness in defining necrosis subtype”, and it leads to the following detailed descriptions “may lead to confusion of their concepts in research fields and would disturb their clinical applications”.
In the text in several places the genes and proteins are mixed. The best example is the sentence in line 146 “genes were identified… including ribosomal protein L8…” should be including genes encoding ribosomal protein L8, iron response….etc.
>>> We modified the sentence following the suggestion of reviewer 1. Also abbreviation of gene names are indicated by Italic.
I don't like the use of the term homology to refer to the process, a term used several times by the authors. I don't seem to understand it.
>>> Pathological process or process is defined as a term by NIH National Library of Medicine (https://www.ncbi.nlm.nih.gov/medgen/18325#:~:text=Definition,from%20NCI%5D), and widely used as a medical and biological term.
What are circles, triangles and crosses in Table 2 and 3. I don’t understand the tables.
>>> We described the meaning of circles, triangles and crosses in legends of Table 2 and 3.
The sentence in lines 258-262 is too long and difficult to understand, it should be simplified. Please also clarify the relationship between HMGB1 and cell death.
>>> We divided the sentence and simplified it. We also added description about the relationship between HMGB1 and cell death.
The information about HMGB1 itself and its role in inflammation is missing.
>>> The previous version included a section named as “HMGB1 released from necrotic neurons induces neuroinflammation”. In addition to this, we newly added information about HMGB1 at the first site where HMGB1 is referred.
There are some typos (lines 55, 84), unnecessary underlying like in lines 33-34.
>>> We corrected the errors.
Reviewer 2 Report
Comments and Suggestions for Authors
The authors did a very good job summarizing the topic and covering relatively new ideas related to TRIAD. Here are some additional suggestions that might be helpful for readers with a more diverse scientific background.
-
Combining Figures and Adding a Summary Table:
- Consider merging Figure-1 and Figure-2 to create a more cohesive visual representation.
- Introduce a summary table detailing disease names, types of cell death reported, investigated areas/cell types (in vivo/cell culture), remarks, and references for clarity and quick reference.
-
Incorporating Cellular Pathway Summary:
- Integrate a figure summarizing cellular pathways mentioned in Section 3, focusing on types of necrosis and apoptosis. This visual aid will aid comprehension for general readers.
-
Including Insights on Parkinson's Disease (PD):
- Expand on Parkinson's disease and the death of dopaminergic neurons. Despite being highly prevalent, PD isn't extensively covered. Provide additional insights into the mechanisms specific to PD neurodegeneration.
-
Clarifying Inflammation Connection:
- Enhance clarity on the connection between neurodegeneration and inflammation by incorporating a figure illustrating the interplay between these two systems.
-
Providing Detailed Experimental Methods for Figure-3:
- Provide a detailed experimental methodology for the original data presented in Figure-3. Clarify whether similar findings hold true for other mice models of Alzheimer's disease (e.g., 3xTg).
-
Elaborating on PANTHOS and PAAS Concepts:
- Add more details regarding PANTHOS and PAAS, and provide comprehensive references while introducing these concepts to enhance readers' understanding.
-
Addressing Mechanisms Triggering Cell Death:
- Consider adding a section or insights on mechanisms that trigger cell death in different neurodegenerative diseases. Discuss common patterns, such as aggregated proteins that lead to cell death. Provide references or direct readers to relevant resources discussing these mechanisms.
- Lastly, the review starts by looking into all the necrotic pathways but gradually focuses on TRIAD (especially in the conclusion and somewhat in the inflammation section). If possible, please include some more insights related to other cell death mechanisms in these sections.
NA
Author Response
Reviewer 2
The authors did a very good job summarizing the topic and covering relatively new ideas related to TRIAD. Here are some additional suggestions that might be helpful for readers with a more diverse scientific background.
>>> We appreciate very much these thoughtful advices from reviewer 2. We completely understand that suggested changes would improve the manuscript substantially. We followed some advices as requested, but we are afraid some of them would be difficult due to the current situation of knowledge in the field or due to the limited focus of this review paper.
- Combining Figures and Adding a Summary Table:
Consider merging Figure-1 and Figure-2 to create a more cohesive visual representation.
>>> We followed the advice and merged previous Figure 1 and 2.
Introduce a summary table detailing disease names, types of cell death reported, investigated areas/cell types (in vivo/cell culture), remarks, and references for clarity and quick reference.
>>> We followed the advice and added the summary table.
- Incorporating Cellular Pathway Summary:
Integrate a figure summarizing cellular pathways mentioned in Section 3, focusing on types of necrosis and apoptosis. This visual aid will aid comprehension for general readers.
>>> We added a figure summarizing cellular pathways, following the advice.
- Including Insights on Parkinson's Disease (PD):
Expand on Parkinson's disease and the death of dopaminergic neurons. Despite being highly prevalent, PD isn't extensively covered. Provide additional insights into the mechanisms specific to PD neurodegeneration.
>>> We added a section for the involvement of TRIAD and other necrosis subtypes in Parkinson’s disease.
- Clarifying Inflammation Connection:
Enhance clarity on the connection between neurodegeneration and inflammation by incorporating a figure illustrating the interplay between these two systems.
>>> We added a figure following the advice.
- Providing Detailed Experimental Methods for Figure-3:
Provide a detailed experimental methodology for the original data presented in Figure-3. Clarify whether similar findings hold true for other mice models of Alzheimer's disease (e.g., 3xTg).
>>> This figure is already published in ref 48, and details of experimental methods were described in that paper. Therefore, we referred the paper in caption. We did not examine 3xTg mice by our hands.
- Elaborating on PANTHOS and PAAS Concepts:
Add more details regarding PANTHOS and PAAS, and provide comprehensive references while introducing these concepts to enhance readers' understanding.
>>> We added details of PANTHOS and PAAS, following the advice.
- Addressing Mechanisms Triggering Cell Death:
Consider adding a section or insights on mechanisms that trigger cell death in different neurodegenerative diseases. Discuss common patterns, such as aggregated proteins that lead to cell death. Provide references or direct readers to relevant resources discussing these mechanisms.
>>> We added a short section about the relationship between protein aggregation state and timing of necrosis triggering.
Lastly, the review starts by looking into all the necrotic pathways but gradually focuses on TRIAD (especially in the conclusion and somewhat in the inflammation section). If possible, please include some more insights related to other cell death mechanisms in these sections.
>>> We responded to this line of advice, partially in the response to the comment 3 (3. Including Insights on Parkinson's Disease). We appreciate this advice, but adding an enormous amount of insights to other necrosis subtypes is beyond the scope of this article, and we think it will be performed in the next review article.

Reviewer 3 Report
Comments and Suggestions for Authors
The manuscript discusses the evolving understanding of neuronal cell death mechanisms in neurodegenerative diseases, highlighting recent advances that challenge the traditional view of apoptosis as the sole mechanism. The review delves into emerging subtypes of necrotic neuronal cell death, offering an updated summary and exploring their potential roles in neurodegenerative processes. Among the various necrosis subtypes, including necroptosis, paraptosis, ferroptosis, and pyroptosis, the review singles out transcriptional repression-induced atypical cell death (TRIAD) as a noteworthy mechanism. TRIAD is associated with the functional deficiency of TEAD-YAP and self-amplifies through the release of HMGB1. The study suggests TRIAD as a plausible mechanism of neuronal cell death in Alzheimer's disease and other neurodegenerative conditions. Furthermore, the manuscript underscores the dual impact of TRIAD, not only inducing cell death but also triggering brain inflammatory responses through HMGB1 release. This connection between neurodegeneration and neuroinflammation is proposed as a potential link that warrants further investigation. However, authors missed another important phenomenon. The neurodegeneration-induced proliferation (see: https://pubmed.ncbi.nlm.nih.gov/24700150/ ; DOI:10.3969/j.issn.1673-5374.2012.14.001 ) has been previously reported and degeneration-induced induced postnatal neurogenesis has an enormous therapeutic potential and needs further exploration. Overall, the manuscript provides an insightful overview of the current state of knowledge regarding neuronal cell death mechanisms in neurodegenerative diseases and introduces TRIAD as a significant player in this context.
Comments on the Quality of English LanguageN/A
Author Response
Reviewer 2
The manuscript discusses the evolving understanding of neuronal cell death mechanisms in neurodegenerative diseases, highlighting recent advances that challenge the traditional view of apoptosis as the sole mechanism. The review delves into emerging subtypes of necrotic neuronal cell death, offering an updated summary and exploring their potential roles in neurodegenerative processes. Among the various necrosis subtypes, including necroptosis, paraptosis, ferroptosis, and pyroptosis, the review singles out transcriptional repression-induced atypical cell death (TRIAD) as a noteworthy mechanism. TRIAD is associated with the functional deficiency of TEAD-YAP and self-amplifies through the release of HMGB1. The study suggests TRIAD as a plausible mechanism of neuronal cell death in Alzheimer's disease and other neurodegenerative conditions. Furthermore, the manuscript underscores the dual impact of TRIAD, not only inducing cell death but also triggering brain inflammatory responses through HMGB1 release. This connection between neurodegeneration and neuroinflammation is proposed as a potential link that warrants further investigation.
>>> Thank you for your kind evaluation of our manuscript.
However, authors missed another important phenomenon. The neurodegeneration-induced proliferation (see: https://pubmed.ncbi.nlm.nih.gov/24700150/ ; DOI:10.3969/j.issn.1673-5374.2012.14.001 ) has been previously reported and degeneration-induced induced postnatal neurogenesis has an enormous therapeutic potential and needs further exploration.
>>> We agree with this opinion of reviewer 2, and appreciate very much introducing the important paper. However, this topic is beyond the scope of this paper, and it needs another review paper.
Overall, the manuscript provides an insightful overview of the current state of knowledge regarding neuronal cell death mechanisms in neurodegenerative diseases and introduces TRIAD as a significant player in this context.
>>> Thank you very much again for the kind evaluation.
Round 2
Reviewer 1 Report
Comments and Suggestions for Authors
Please find comments in attached file.

Author Response
Reviewer 1
In manuscript entitled “Necrosis links neurodegeneration and neuroinflammation in neurodegenerative disease” Hidenori Homma et al., authors did improvement, but I have still some comments.
>>> We appreciate detailed comments from the reviewer that were very useful to improve the manuscript.
The list is belove.
Title of chapter 2 – I propose: Variety of necrosis types: …
>>> We changed as requested.
Line 103, 184, 194, 216 - I still have a problem with understanding the word homology in relation to processes. Homology in biology refers to structures, genes, proteins, having a common ancestor. Should I understand that subtypes of the necrosis process are variants of one and the same process? Or, does the term refer to similarities? I need an explanation to clarify what is meant. Please also clarify this in a text.
>>> We are little bit puzzled which definition the reviewer uses. We agree that for instance in genetics “homology” is used to mean the same ancestor. However, although there seem to be some discussions in terminology, the term “homology” is more widely used in general. Here, we used “similarity” as 100% identity in a parameter and “homology” as less than 100 percentage of identity in parameter. We clarified it in the text as requested.
Line 107 transcriptional upregulation is of gene expression so it should be: “upregulation of expression of genes encoding caspase-9... “
>>> We changed as requested.
Line 115 What is a morphological definition? Do you mean a definition based on the observation of cell morphology? What do you mean "with morphological description"? - do you mean based on morphological description?
>>> biological definition of a phenomena should be done with multiple factors, such as morphological characteristics, biochemical characteristics, cell signaling characteristics, and maybe genetic characteristics if necessary. Here, we meant biological side of definition (not total criteria of definition) with “morphological definition”.
Line 152 IREB2 in italics as it is a name of the gene
>>> We corrected.
Line 163 give the full name of TRIAD, here it is for the first time in the text, the summary does not count.
>>> We are confused with this comment. TRIAD appears in line 84 firstly in the main text. However, we did not describe the full name in line 84 because we wrote it already in Summary (Abstract). In some journals like Nature, full name should be appear in Abstract if it is the firstly appeared position, and afterwards no need to rewrite in the main text. We follow the rule of this journal IJMS.
Lines 165, 202, 211 and in Table 1 - I think I am still not understood. What I meant when I asked you not to write "KDEL monitoring" is that the sequence of the amino acid residues cannot be monitored. What is monitored is the localisation of a fusion protein consisting of a fluorescent protein with the aforementioned sequence. Since the KDEL sequence determines the retention of the protein in the ER, this is used to visualise the ER by observation of localization of fusion protein by fluorescence microscopy. At least I use such a fusion for this purpose. It is not enough to write ECFP-KDEL instead of KDEL. It should be written ; monitored by observation of ECFP-KDEL fusion protein localisation.
>>> We are not so sure about we are understanding what the reviewer requested, but we added some words according to the words of the reviewer in line 165, where this kind of description firstly appears.
Table 1 What is MLKL – there is no information in a text
>>> We added the information in Note of Table 1
Line 138, 296 and Table 1 - Ras mutant cancer cells or RAS genes mutated – depending if you think about abnormal protein or mutated gene.
>>> We changed them to RAS (Italic) mutated cancer cells.
The title of Table 2 - I propose changing of words order to be: Criteria used for simultaneous analysis of necrosis subtypes in AD and FTLD…”, as analysis is of necrosis subtypes.
>>> We followed the recommendation.
Table 2 and 3 - Double titles are not necessary; the description of circles, triangles and cross should be in a table description not in the text.
>>> We corrected them.
Line 277 to ischemic brain I still have a problem with Figure 3. If the figure is published than this should be stated in caption. If not and authors decide to leave it here than the caption has to be more explanatory. The symbols in the graph are too small and even on a computer it is impossible to distinguish them, especially that are located on an axis. The solution is to put them belove the axis. Also descriptions p Ser46-MACKS in red have to be better visible.
>>> It was already published (ref 48), and we state the reference in caption.
Lines 287-288 this is a text which should be a part of the caption. In caption to this figure also should be written that AD model mice strains (5xFAD and APP-KI) and in control B6 strain.
>>> We write these sentences as caption for Figure 3, but the automatic conversion from doc to PDF of this journal submission system made the mistake in the position. We added strains of these mice in the caption.
Lines 268, 276, 290 Table 1 - Unify the way you describe phosphorylated MARCKS. The first time, specify the phosphorylated residue and define how you will name the phosphorylated MARCKS. Later, use this introduced name.
>>> We followed the comment. We also followed the rule in caption for Figure 3.
Lines 273- 274 To avoid using the same word leading and give more explanation I suggest to write: resulting in inactivation of Ku70, a protein which function in DNA repair and maintenance. This leads to DNA damage …
>>> We appreciate the comment very much. However, kinases that mediate signaling to neurite degeneration and cell death are different in the signaling pathways from TLR4 triggered by HMGB1. We used some key phase suggested and modified these sentences.
Line 323- Simplify the sentence by omitting the repetition of the name YAP in it
>>> We followed.
Line 326 Are references [63-65] in a good place? For me should be in the end of the sentence.
>>> We corrected it.
Line 332 I still don’t know what question has to be examined? Above are statement sentences. Do you mean: It is known that ROS induces ferroptosis [40], and since ROS levels increase after TLR4 induction, so the question is whether ferroptosis can also be induced in this case. If so clarify this.
>>> Exactly. We employed the phase of the reviewer and clarify it.
Lines 335- 339 Not to repeat what is already in lines 155-156 and to shorten sentence I propose: Following the initial discovery of alpha-amanitin-induced TRIAD necrosis in cultured primary neurons influenced by the presence of different YAP isoforms [41], a similar phenotype mediated by YAP inactivation has been detected in patients with Huntington's disease [43] and in mouse models of the disease. YAP inactivation has been found to occur through interaction with the abnormal huntingtin protein (Htt) and is associated with disease pathology [44].
>>> We employed the phase of the reviewer.
Line 371 description of what is marked red, purple and green in Figure 4 should be in the figure caption.
>>> This part should be in caption as reviewer said. However automatic transfer from doc to PDF file made the error. Please have a look at the submitted doc file. We can also correct this error in the proof step.
Reviewer 2 Report
Comments and Suggestions for Authors
Great work with the incorporations. The table and the pathway figure present the ideas nicely and in an easy-to-identify way. The section on PD also adds valuable information.
couple of doubts,
1. Is PANTHOS also an acronym? If so, please include it.
2. Please explain what it means by secondary and tertiary necrotic neurons in Figure 6.
Comments on the Quality of English LanguageNA
Author Response
Great work with the incorporations. The table and the pathway figure present the ideas nicely and in an easy-to-identify way. The section on PD also adds valuable information.
>>> Thank you for your kind evaluation.
couple of doubts,
- Is PANTHOS also an acronym? If so, please include it.
>>> We referred the meaning of PATHOS according to the original report.
- Please explain what it means by secondary and tertiary necrotic neurons in Figure 6.
>>> We explained it in Figure 6 legend.